# Radiation Retinopathy: Microangiopathy-Inflammation-Neurodegeneration

**DOI:** 10.3390/cells14040298

**Published:** 2025-02-18

**Authors:** Anja-Maria Davids, Inga-Marie Pompös, Norbert Kociok, Jens Heufelder, Sergej Skosyrski, Nadine Reichhart, Antonia M. Joussen, Susanne A. Wolf

**Affiliations:** 1Department of Ophthalmology, Charité–University Medicine Berlin, Augustenburger Platz 1, 13353 Berlin, Germanyinga.pompoes@charite.de (I.-M.P.); norbert.kociok@charite.de (N.K.); antonia.joussen@charite.de (A.M.J.); 2Berlin Protonen am Helmholtz-Zentrum Berlin, Charité–University Medicine Berlin, 14109 Berlin, Germany; jens.heufelder@charite.de; 3Max-Delbrück-Center for Molecular Medicine in the Helmholtz Society, Department of Psychoneuroimmunology, Robert-Rössle-Str. 10, 13025 Berlin, Germany

**Keywords:** retinopathy, radiation, retina

## Abstract

Purpose: Proton irradiation is used to treat choroidal melanoma of the eye. The impact on non-malignant retinal cells is currently understudied. Therefore, we here report a mouse model to investigate the impact of proton irradiation on the retina. Methods: We performed a proton beam irradiation of 5–15 Cobalt-Gray-Equivalent (CGE) of the eyes of female C57Bl6/J (Cx3cr1^+/+^), Cx3cr1^gfp/+^ and Cx3cr1^gfp/gfp^ mice mimicking the clinical situation and evaluated the structure, function and cellular composition of the retina up to 24 weeks after irradiation. Results: Proton beam irradiation of the eye with 15 CGE leads to cataract formation after 24 weeks without affecting the gross anatomy of the retinal vasculature as shown by Fundus imaging in all genotypes respectively. However, 10 and 15 CGE, lead to a significant decrease in NG2 positive cell numbers and all three dosages induced an increase in GFAP immunoreactivity. At 24 weeks a dosage of 15 CGE resulted in functional impairment and a decrease of NG2 positive cells in both WT and Cx3cr1 animals. Iba1 cell immunoreactivity was increased in all genotypes. However, in the Cx3cr1 animals the invasion of Iba1 cells into the deep vascular layer was partially prevented. This was accompanied by a less severe functional impairment in the irradiated Cx3cr1^gfp/gfp^ vs. WT. Conclusions: Although the gross anatomy of the retina does not seem to be affected by proton beam irradiation, the cellular composition and retinal function changed significantly in both WT and Cx3cr1 mice reflecting the clinical situation. Moreover, cataract formation was one of the major long-term effects of irradiation. We conclude that the murine model (WT and Cx3cr1 genotype) can be used to investigate proton-beam associated side effects in vivo as well as to test prospective interventions. Moreover, the loss of Cx3cr1 seems to be partially protective.

## 1. Introduction

Irradiation of the ocular globe is the first line treatment in patients who are suffering from a choroidal melanoma. Since these are small tumor findings, only a few millimeters in size, a focused radiation with minimizing damage to surrounding tissues is desirable.

In everyday clinical practice, plaque brachytherapy with, for example, Ruthenium-106 (beta radiation), Iodine-125 and Palladium-103 (both photon radiation) plaques and, where appropriate facilities are available, external beam radiotherapy with charged particles such as protons have become established [1]. A targeted irradiation of the choroidal melanoma with proton beams allows an eye-preserving therapy [2,3,4]. However, there is still a risk of developing a radiation retinopathy after ocular radiation, which can lead to a loss of function and, in rare cases, as well to an enucleation [5]. Radiation retinopathy has a negative impact on the visual outcome of proton irradiation of choroidal melanoma [6,7].

In the course of clinical observations as well as studies, it was revealed that radiation retinopathy is a slowly progressive microangiopathy of the retinal vessels that becomes evident after a delay of approximately 2 years after proton irradiation [8]. Various vascular pathologies can be observed, including ischemic or proliferative retinopathy and maculopathy [1,9,10]. Based on these observations, there could be pathophysiological similarities to diabetic retinopathy, which is characterized by a microangiopathy, leading to progressive capillary nonperfusion [11], but also an inflammatory component.

Central to the development of Radiation retinopathy is the dysfunction of endothelial cells, which line the interior surfaces of blood vessels and are pivotal in maintaining vascular integrity and the blood-retinal barrier. Radiation exposure disrupts these cells, leading to increased vascular permeability, loss of cell-cell contact, and subsequent cell death. These alterations can precipitate a cascade of pathophysiological events in the retina, including edema and neovascularization, which are characteristic of radiation retinopathy [12].

The damage to endothelial cells from radiation is mediated primarily through oxidative stress and an inflammatory response. Oxidative stress, induced by radiation, results in the production of reactive oxygen species (ROS) that can damage cellular proteins, lipids, and DNA. Concurrently, radiation can trigger a robust inflammatory response characterized by the upregulation of cytokines and adhesion molecules, fostering an environment that facilitates leukocyte adhesion and migration into retinal tissues. These processes exacerbate endothelial injury and contribute to the vascular anomalies observed in radiation retinopathy [13,14].

Insights into these mechanisms have been greatly advanced by studies utilizing mouse models, which have been instrumental in elucidating the cellular and molecular underpinnings of radiation retinopathy. These models have highlighted significant changes in retinal vascular architecture following radiation exposure, including vessel dilation, tortuosity, and the formation of microaneurysms. Furthermore, mouse models have served as critical platforms for testing the efficacy of therapeutic interventions, such as anti-VEGF therapies, which aim to mitigate endothelial damage and preserve retinal function [15].

Parallel clinical observations in humans have detailed a spectrum of microvascular abnormalities in radiation retinopathy, ranging from cotton wool spots to severe hemorrhages and neovascularization. The management of these clinical manifestations primarily involves the administration of anti-VEGF injections, corticosteroids, and in some cases, laser photocoagulation, highlighting the need for continued research into more effective therapeutic strategies [16].

Recent advancements in this field are focusing on identifying early biomarkers for radiation retinopathy and developing novel protective agents that can shield endothelial cells from radiation-induced harm.

To date, there are no experimental studies using proton irradiation on mouse eyes. But in experiments with external beam radiation from rat eyes, including protons, endothelial cell loss, capillary dropout, vessel occlusion, retinal smooth muscle loss, retinal degradation with photoreceptor loss were observed [1].

Most animal model studies focus on the vasculopathy of radiation retinopathy [1]. Less is known about inflammatory mechanisms, but they also appear to play a significant role. Focusing on the inflammatory components of radiation retinopathy, such as microglia activation, and the roles of glial fibrillary acidic protein (GFAP), vimentin, and pericytes, is now increasingly important. While much of the research has traditionally concentrated on endothelial cell dysfunction, the broader inflammatory response plays a crucial role in the progression and severity of the disease. Microglia, the resident immune cells of the central nervous system, become activated in response to radiation and can exacerbate damage through the release of pro-inflammatory cytokines and reactive oxygen species. Similarly, upregulation of GFAP and vimentin in astrocytes and Müller cells, respectively, indicates reactive gliosis, which contributes to the inflammatory environment and structural disorganization of the retina. Pericytes, vital for capillary stability, undergo changes that further disrupt vascular integrity and blood-retinal barrier function. By shifting the focus toward these elements, researchers can uncover new therapeutic targets that address the complex network of cellular interactions and inflammatory pathways involved in radiation retinopathy, potentially leading to more effective treatments that mitigate the broader spectrum of pathological changes beyond just the endothelial cells. In a publication on radiation retinopathy in rat eyes following irradiation with X-rays, they showed an invasion of microglia and macrophages into the retinal pigment epithelium and that this observation is related to an outer blood retina barrier disruption [17].

The aim of this study, therefore, was to establish a model of radiation retinopathy following proton beam irradiation (as in everyday clinical practice with humans) in mice and to investigate irradiation-related morphological and functional consequences in this model focusing on the inflammatory components such as microglia. Therefore, transgenic Cx3cr1 mice were included in the experiments. The homozygote Cx3cr1^gfp/gfp^ mice lack the fractalkine receptor, which is mainly expressed by microglia in the brain and retina and has an important role in trafficking microglia cells to and from the subretinal space [18].

A possible effect of the altered microglia cell migration on the development of radiation retinopathy could be a new therapeutic approach in preserving visual function after proton irradiation of the ocular globe.

## 2. Methods

### 2.1. Study Approval

All animal experiments were committed in conformity with the German law of animal protection and the National Institute of Health guidelines for care and use of laboratory animals. All experiments were approved by the local ethics committee on animal research (LaGeSo, Landesamt für Gesundheit und Soziales Berlin, Germany, G0218/13). The experimental design and execution follow the ARVO Statement for the Use of Animals in Ophthalmic and Vision Research.

### 2.2. Animal Model

C57/Bl6J mice were purchased from Janvier (Cedex, France) and were used to establish the irradiation protocol as well as to perform a first characterization of proton radiation-related anatomical and functional ophthalmological changes.

Subsequently, transgenic Cx3cr1^gfp/gfp^ mice on a C57BL/6J background were used to study the impact of Cx3cr1 signaling on the ophthalmological effects of proton radiation. In this mouse line, the first 390bp of the Cx3cr1 gene are replaced by the gene encoding the enhanced green fluorescent protein (EGFP) and therefore the homozygote mice lack the fractalkine receptor, which is mainly expressed by microglia in the brain and retina [18,19]. Heterozygote mice of this strain are often used as reporter mice to visualize microglia in vivo and ex vivo by EGFP fluorescence. Consequently, it is assumed that in heterozygote mice Cx3cr1 signaling is possible without restriction. However, microglia in Cx3cr1^gfp/+^ mice exhibit an intermediate transcriptional profile between wild-type microglia and microglia in the Cx3cr1^gfp/gfp^ mice [19,20].

Therefore, we included this strain into our study to test whether the heterozygote mice (Cx3cr1^gfp/+^)compared to wild type mice (Cx3cr1^+/+^) and to investigate the impact of Cx3xr1 loss on the outcome in the homozygote mice (Cx3cr1^gfp/gfp^). The Cx3cr1 strain was kindly provided by F. Sennlaub (Department of Therapeutics, Sorbonne Université, INSERM, CNRS, Institute de la Vision, Paris, France) and tested negative for RD8 mutation. In our local animal facility, the breeding was sustained using the C57BL/6J line. Independent of the genotype, all mice were female, 10–12 weeks of age and weighing 20 to 25 g at the beginning of the experiments to prevent bias due to inter-individual variability.

### 2.3. Irradiation Procedure

The proton irradiation was performed at the Helmholtz-Zentrum Berlin für Materialien und Energie (HZB) in Berlin-Wannsee. The proton facility is primarily designed and built to treat patients, who suffer from choroidal melanoma [21,22]. The responsible radiation physicists on site created the plan for the irradiation of the mouse eyes in a blinded fashion for the genotype. Irradiation field and aperture were adapted to the murine anatomy. The mice were anesthetized and placed onto a customized platform in front of the clinical beam line. The anesthesia was done by an intraperitoneal injection of a dose of 100 mg/kg BW Ketamine and 12 mg/kg BW Xylazine. After irradiation, the mice were brought back to their cage to wake up. One eye was irradiated while the contralateral eye and both eyes of the sham animals served as control. Sham animals were only placed in front of the clinical beam line for 5 min to control for the stress without irradiating the animals.

The data on the radiation field include the aperture diameter, which measured 9 mm, the depth of penetration into the mouse tissue (range), which was 5mm, and the length of the does plateau (modulation) measuring 5 mm. The dose rate was approximately 10.5 Gy/min or 11.6 Cobalt Gray Equivalent (CGE)/min.

The following doses were applied for establishing a radiation dose-response curve for mice: 4.55 Gy or 5.0 CGE, 9.09 Gy or 10.0 CGE and 13.64 Gy or 15.0 CGE. The doses were determined by calculating the equivalent mouse dosages based on those used in patients. A detectable effect was observed at 13.64 Gy 15 GDE, which was subsequently chosen for further experiments.

### 2.4. Scanning Laser Ophthalmoscopy

The in vivo imaging of the retina of both mouse eyes was performed using a scanning laser ophthalmoscope at 8, 16 and 24 weeks after proton irradiation as described previously [23].

Briefly, the mice were anesthetized prior to the examination. The corneas were moisturized with Hylo-Vision (NaCl 0.1%; Omnivision, Puchheim, Germany) and the mice were placed on a customized platform in front of the device. Spectralis HRA-OCT (Heidelberg Engineering, Heidelberg, Germany) was settled with a c-curve of 3.5 and incorporated a wide-angle lens 55°.

Fluorescence angiography (FA) was performed to assess the vessel structure and any leakage. For this purpose, the mouse eyes were examined with the FA mode (diode laser, wavelength of 488 nm), 5 min after the subcutaneous injection of fluorescein (5 mg/kg, fluorescein 10%; Alcon, Freiburg, Germany).

The auto-fluorescence mode in the ophthalmoscope (AF, 488 nm) was used to display the EFGP positive cells of the heterozygous Cx3cr1^gfp/+^ and homozygous Cx3cr1^gfp/gfp^ mice.

Images of the inner and outer retina were taken centrally and peripherally and were analyzed using the Heidelberg Eye Explorer 1.7.0.0 software.

### 2.5. ERG

The retinal function was measured with the help of Ganzfeld electroretinography (ERG) as described previously [24].

Briefly, before the measurement, the mice were adapted to the dark overnight. Then, after the mice were anesthetized with xylazine (15 mg/kg body weight) and ketamine (100 mg/kg) and the pupils dilated with 0.5% tropicamide and 2.5% phenylephrine hydrochloride, they were placed on a warm plate in a Ganzfeld bowl (Roland Consult, Brandenburg, Germany).

The ERG was recorded with monopolar contact lens electrodes. Platin needles, which were fixed subcutaneously, served as reference and ground electrodes.

The examination took place 3 days and 2, 8, 16 and 24 weeks after proton irradiation.

### 2.6. Immunohistochemistry

The eyes for the immunofluorescence staining were obtained 24 weeks after proton irradiation. To visualize GFAP and Vimentin to examine gliosis and fibrosis, the enucleated eyes were fixed in 4% PFA overnight and embedded in paraffin to make sections from the central bulbus with a thickness of 4 µm. After the sections were deparaffinized and rehydrated, they were incubated with Proteinase K for antigen retrieval and permeabilized in TBS with 0.5% Triton X-100. After 1 h blocking with 5% BSA, the sections were incubated overnight with the primary antibodies (guinea pig anti-GFAP, Synaptic Systems, Göttingen, Germany, 173004, 1:100, chicken anti-Vimentin, Novus Biologicals, Centennial, CO, USA, NB300-223, 1:50) and the day after with DAPI (Roche, Basel, Switzerland, 10236276001, 1:100) and the secondary antibodies (Alexa Fluor 568 goat anti-guinea pig, A11075, 1:8000, Alexa Fluor 488 donkey anti-chicken, Jackson, West Grove, PA, USA, 711-545-152, 1:4000) for one hour. Covered by glass slides, sections were visualized with an Axio Imager M2 fluorescence microscope (ZEN-Software 3.1 Blue Edition, Zeiss, Oberkochen, Germany). The 488 and 568 channel images were analyzed separately with Image j 1.53a (Wayne Rasband, National Institute of Health, USA) and the retinal area was defined excluding the photoreceptor outer segments. Finally, the integrated density, as sum of values of the pixels in the image or selection equivalent to the product of area and mean gray value, was measured of a defined retinal area excluding the photoreceptor outer segments.

To further investigate changes in the vascular component on a cellular level, we employed isolectin B4 (IB4) for general vascular labeling and NG2 for specific pericyte identification in retinal flat mounts. Isolectin B4 is a lectin isolated from the seeds of the tropical African legume Griffonia simplicifolia, which specifically binds to carbohydrate structures, primarily N-acetyld-galactosamine end groups. It is widely utilized for staining murine endothelial cells to visualize and analyze vascular structures. However, Isolectin B4 also exhibits cross-reactivity with microglia [25]. Neuroglial antigen 2 (NG2), a chondroitin sulfate proteoglycan, is predominantly expressed on the surface of oligodendrocyte precursor cells and pericytes. In the context of vascular biology, NG2 plays a crucial role in vascular stability and angiogenesis. Antibodies against NG2 are commonly used as pericyte markers in retinal vessels, facilitating the quantification of the pericyte/endothelial cell ratio [26,27]. In this experimental study, the number of pericytes was analyzed in relation to the length of Isolectin B4-positive retinal vessels. We have used the Isolectin B4 staining to visualize vessels in toto with no specific focus on endothelial cells.

To count microglia, cells were labeled with Iba1 in retinal flat mounts. Iba1 (ionized calcium-binding adaptor molecule 1) is specifically expressed in microglia and macrophages and is widely used as a marker to identify and study these cell types in the retina [28].

For this purpose, the eyes were fixed in 4% PFA for 15 min. Subsequently, the cornea was removed via a circumferential limbal incision, lens and vitreous were removed and four radial incisions were made, to achieve a flattening of the bulbus. Finally, the retina was dissected from the choroid. Retinal flat mounts were transferred into a 24-well plate and permeabilized with 5% Triton X-100 in TBS overnight. The flat mounts were incubated with Isolectin-IB4 (either GS Isolectin-B4, Alexa Fluor 488-conjugated, Invitrogen, I21411, 1:200 or Alexa Fluor 647-conjugated, Invitrogen, Carlsbad, California, USA, I32450, 1:200) to visualize the vasculature. In samples treated with the 488-conjugated isolectine, the primary antibody (rabbit anti-NG2 Chondroitin Sulfate Proteoglycan, CY3-conjugated, Sigma-Aldrich, Burlington, Massachusetts, USA, AB5320C3, 1:200) was applied overnight, after one hour blocking with 15% serum in TBS. NG2 labeled cells were counted in images from the inner and outer vessel layer using an Axio Imager M2 fluorescence microscope (ZEN-Software 3.1 Blue Edition, Zeiss, Oberkochen, Germany). Samples treated with the 647-conjugated Isolectin, were incubated with a primary antibody for Iba1 (rabbit anti-Iba1, Abcam, Cambridge, UK, ab178847, 1:200) overnight followed by 2 h incubation with a secondary antibody (Alexa Fluor 488 donkey anti-rabbit, Dianova, Hamburg, Germany, 711-545-152, 1:1000) before the samples were mounted on glass slides. To count Iba1 positive cells the covered flat mounts were scanned by using the z-stack mode at a confocal Leica SPE microscope (Leica Microsystems GmbH, Wetzlar, Germany).

### 2.7. FACS

Single cell suspensions were obtained from the retinas by mechanical dissociation and cell separation through a moist cell sieve with pores of 70 µm in size. Cells were labeled with CD11b Monoclonal Antibody (M1/70, PE-Cyanine7, eBioscience™, San Diego, CA, USA, 25-0112-82) and CD45 eflour 450 (30-F11, eBioscience™, San Diego, USA, 48-0451-82) for 15 min in FACS buffer (PBS and 1% FCS) on ice in the dark and immediately analyzed using a FACS Canto II (BD, Franklin Lakes, NJ, USA) and the FlowJo v10.10 software as described elsewhere [29].

### 2.8. Data Analysis

The Shapiro-Wilk-Test was used to evaluate the normal distribution of the variables and the Levene Test to assess the equality of variances. In case of normally distributed variables the paired samples *t*-test was performed to compare the variables. One-way (one independent variable) or two-way (two independent variables) ANOVA was used when comparing more than two groups. The Bonferroni-Test was used as post hoc test. For non-normally distributed variables non-parametric tests were used: the Wilcoxon-Test for two dependent variables, the Mann-Whitney-U-Test for two independent variables, the Friedmann-Test for multiple dependent variables and the Kruskal-Wallis-Test for multiple independent variables. *p*-values lower than 0.05 were determined to be significant. In case of missing data, a pairwise deletion was made.

The data analysis was performed with the program SPSS Statistics (Version 23.0 for Mac, Chicago, IL, USA). For the graphical representation of the data box-and-whisker plots (whiskers represent the minimum and the maximum value of the data set) or bar charts (mean ± SEM) were used. The single data points were superimposed in both plots.

## 3. Results

### 3.1. Cataract Development in Wild Type Animals

All eyes irradiated with 10 and 15 CGE had developed a dense cataract 24 weeks after proton irradiation (Figure 1D). Consequently, no fundus imaging of the affected eyes was possible at this time point. This also applied to 2 of the 6 eyes that were irradiated with 5 CGE at 24 weeks.

Sixteen weeks after proton therapy, a fundus imaging was possible in all eyes in the group of animals irradiated with 5 CGE. Two out of six eyes irradiated with 10 CGE and in 4 out of 6 eyes in the group of animals irradiated with 15 CGE developed a cataract (Figure 1D). At 8 weeks a fundus imaging was possible in all eyes of all three irradiation dosages. We did not detect any cataract development in the control and sham eyes at any time point.

There was no hint to abnormalities in the performed fluorescence angiographies (Figure 1A). Neither vessel tortuosity, nor increase in the vessel diameter or leakage could be detected. In the fundus autofluorescence images no changes or abnormalities were detectable either (Appendix A). Despite the lack of vessel abnormalities, the number of cataracts already points towards a detrimental effect of higher radiation dosage.

### 3.2. Vascular Component and Gliosis/Fibrosis in Wild Type Animals

In the retina of the eyes which were irradiated with 10 and 15 CGE, we found a significantly reduced number of pericytes per mm vessel length in the superficial vascular layer compared to the non-irradiated retina (10 CGE irradiated 4.39 ± 1.5 vs. non-irradiated 9.81 ± 1.3, *n* = 4, paired two-sample *t*-test, *p* < 0.001; 15 CGE irradiated 3.66 ± 1.4 vs. non-irradiated 10.61 ± 1.9, *n* = 3, paired two-sample *t*-test, *p* = 0.002; Figure 2B) and the retina of the sham group, respectively (10 CGE irradiated 4.39 ± 1.5 vs. Sham 10.18 ± 1.3, *n* = 3, paired two-sample *t*-test, *p* ≤ 0.001; 15 CGE irradiated 3.66 ± 1.4 vs. Sham 10.18 ± 1.3, *n* = 3, paired two-sample *t*-test, *p* ≤ 0.001; Figure 2B). We did not detect a pericyte dropout for the 5 CGE- group. In addition, there was no difference in the number of pericytes per mm vessel length between the images from the periphery of the retina and those from central areas (Appendix A).

GFAP was significantly increased for all radiation doses compared to the contralateral retina in a dose-dependent manner (5 CGE irradiated 28.64 ± 7.4 vs. non-irradiated 21.94 ± 3.97. *n* = 3, paired two-sample *t*-test, *p* < 0.001; 10 CGE irradiated 38.66 ± 10.1 vs. non-irradiated 29.11 ± 9.7, *n* = 3, paired two-sample *t*-test, *p* = 0.007; 15 CGE irradiated 41.95 ± 8.6 vs. non-irradiated 32.37 ± 8.6, *n* = 3, paired two-sample *t*-test, *p* < 0.001; Figure 3B).

Concerning all radiation dose groups, there was no significant difference between the mean relative integrated density of GFAP in the central and peripheral sections (Figure 3C).

In case of the mean relative integrated density of Vimentin, there was generally no significant difference between irradiated and non-irradiated animals (Figure 3B). This also applies to the images of the central and peripheral area (Figure 3C).

### 3.3. Retinal Function in Wild Type Animals

Retinal function was assessed by performing a Ganzfeld electroretinogram (ERG) examination of both eyes (Figure 4, Appendix A).

Regarding the scotopic a-wave amplitudes at 3 cds/m^2^ corresponding to the photoreceptor activity a descent decrease over time was observed in the irradiated and non-irradiated eyes (Figure 4A). 24 weeks after proton irradiation, the decrease in the irradiated eyes was significantly more pronounced than on the opposite side (irradiated 269.1 µV ± 83.2 vs. non-irradiated 302.6 µV ± 41.8; *n* = 8; paired two-sample *t*-test, *p* > 0.05, Appendix A).

The scotopic b-wave amplitudes at 10 mcds/m^2^ and 3 cds/ m^2^ representing bipolar cell activity were largely stable over time in both the irradiated and non-irradiated eyes (Figure 4B,D). Again, the decrease was significantly more pronounced in the irradiated eyes (irradiated 363.4 µV ± 108.7 vs. non-irradiated 403.4 µV ± 58; *n* = 8; paired two-sample *t*-test, *p* > 0.05, Appendix A; irradiated 505.75 µV ± 156.9 vs. non-irradiated 568.88 µV ± 75; *n* = 8; paired two-sample *t*-test, *p* > 0.05, Appendix A).

The b-a ratio at 3 cds/m^2^ was stable for the entire period up to 24 weeks after proton irradiation (Figure 4C) without a significant difference between the irradiated and non-irradiated eyes (Appendix A). Values above 2.2 reflect an impairment of the rod function and values beneath 1.8 the bipolar cell function.

The scotopic c-waves 250 ms at 18 cd/m^2^, which reflect the function of the pigment epithelium, showed decreasing values from week 16 onwards after radiotherapy for both the irradiated and non-irradiated eyes (Figure 4E). At week 24, the decrease was significantly more pronounced in the irradiated eyes (irradiated 525.6 µV ± 202.9 vs. non-irradiated 596.8 µV ± 64.6; *n* = 8; paired two-sample *t*-test, *p* > 0.05, Appendix A).

The oscillatory potentials at 2.5 cds/m^2^ correspond with the function of the amacrine cells and their interconnections. Regarding this parameter, a decrease over time until finally 24 weeks after proton irradiation was observed (Figure 4G) with no significant difference (Appendix A).

The photopic wave amplitudes at 20cd (background 50 cd/m^2^) correlates with the cone response after 10 min of light adaptation. They were largely stable up to 16 weeks, but clearly reduced at 24 weeks after proton irradiation in irradiated and non-irradiated eyes (Figure 4F). Again, at week 24, the decrease was more pronounced (albeit not significantly) in the irradiated eyes in comparison to the contralateral eyes (irradiated 123.4 µV ± 34.7 vs. non-irradiated 144.3 µV ± 35.3; *n* = 8; paired two-sample *t*-test, *p* = 0.058, Appendix A).

### 3.4. Retinal Function in Cx3cr1 Knock out Animals

To examine the input of Cx3cr1 on radiation retinopathy and evaluate whether these reporter mice are a valid model to study radiation-related effects on the retina, we repeated the previous experiments using transgenic Cx3cr1^gfp/gfp^ and Cx3cr1^gfp/+^. The respective fellow eyes of the irradiated eyes were used as control.

First, a Ganzfeld electroretinogram (ERG) examination of the irradiated and non-irradiated eyes of heterozygous Cx3cr1^gfp/+^ and homozygous Cx3cr1^gfp/gfp^ mice was performed to evaluate the impact of the Cx3cr1 signaling on the retinal function after proton irradiation (Figure 4, Appendix A).

The Scotopic a-waves at 3 cds/m^2^ in response to a single flash of 3 cds/m^2^ showed a decrease over time for both genotypes and was significantly reduced 24 weeks after proton irradiation (Cx3cr1^gfp/+^ irradiated 3 days 370.5 µV ± 3.1 vs. 24 weeks 253.83 µV ± 34.9, *n* = 12, paired two-sample *t*-test, *p* = 0.028; Cx3cr1^gfp/gfp^ irradiated 3 days 390.14 µV ± 52.4 vs. 24 weeks 286.78 µV ± 33.4, *n* = 9, paired two-sample *t*-test, *p* = 0.001, Figure 4A). This decrease was observed in both eyes and thus regarded an ageing effect. Only in case of the eyes of the homozygous Cx3cr1^gfp/gfp^ mice the a-wave amplitudes were significantly more reduced in the irradiated eyes than in the non-irradiated eyes (Cx3cr1^gfp/gfp^ irradiated 287.78 µV ± 33.4 vs. non-irradiated 329.56 µV ± 49.4, *n* = 9, paired two-sample *t*-test, *p* = 0.007, Appendix A) 24 weeks following proton irradiation. No effect of irradiation was detected in WT and Cx3cr1^gfp/+^ heterozygote mice over time in the a-waves.

The scotopic b-wave amplitudes at 10 mcds/m^2^ were significantly decreasing only in the homozygous Cx3cr1^gfp/gfp^ mice at 24 weeks after proton irradiation in both eyes (Cx3cr1^gfp/gfp^ irradiated 3 days 471 µV ± 42.4 vs. 24 weeks 403.67 µV ± 45.4, *n* = 9, paired two-sample *t*-test, *p* = 0.023, Figure 4D) while no ageing effect was visible in WT or Cx3cr1^gfp/+^. There was no significant effect between irradiated and non-irradiated mice regarding the b-wave amplitudes in all genotypes (Appendix A).

The scotopic b-wave amplitudes at 3 cds/m^2^ showed significant decreasing values for both genotypes from week 8 on after proton irradiation (Cx3cr1^gfp/+^ irradiated 3 days 749.25 µV ± 48.8 vs. 24 weeks 544.25 µV ± 67.2, *n* = 12, paired two-sample *t*-test, *p* = 0.023; Cx3cr1^gfp/gfp^ irradiated 3 days 733.86 µV ± 79.7 vs. 24 weeks 566.89 µV ± 69, *n* = 9, paired two-sample *t*-test, *p* = 0.001, Figure 4 B) and were significantly reduced after 24 weeks in the homozygous Cx3cr1^gfp/gfp^ eyes only (Cx3cr1^gfp/gfp^ irradiated 566.89 µV ± 69 vs. non-irradiated 603.44 µV ± 62.1, *n* = 9, paired two-sample *t*-test, *p* = 0.05, Appendix A).

Regarding the b-a ratio at 3 cds/m^2^ a trend to an increase towards 2.2 over time was observed in the irradiated eyes of the heterozygous Cx3cr1^gfp/+^ mice, whereas the values remained stable in the fellow eyes (Figure 4C). Concerning the irradiated eyes of homozygous Cx3cr1^gfp/gfp^ mice, we observed a trend to an increase in the b-a ratio until week 16. However, the value decreased again to 2 24 weeks after proton irradiation with a significant difference in relation to the irradiated eyes of the heterozygous Cx3cr1^gfp/+^ mice (Cx3cr1^gfp/gfp^ irradiated 1.99 ± 0.1 vs. Cx3cr1^gfp/+^ irradiated 2.2 ± 0.2, *n* = 9/12, Mann-Whitney-U, *p* = 0.049, Appendix A).

The scotopic c-wave amplitudes 250ms at 18 cd/m^2^ were decreasing in both eyes of the heterozygous Cx3cr1^gfp/+^ mice significantly after 24 weeks (Cx3cr1^gfp/+^ irradiated 3 days 841.25 µV ± 102.5 vs. 24 weeks 542.58 µV ± 108.3, *n* = 12, paired two-sample *t*-test, *p* = 0.009, Figure 4E). Regarding the homozygous Cx3cr1^gfp/gfp^ mice, they were largely stable over time in both the irradiated and non-irradiated eyes. There was no significant difference between the treated and non-treated side. However, the c-wave amplitudes were significantly higher respectively better in the irradiated eyes of the homozygous Cx3cr1^gfp/gfp^ than in the heterozygous Cx3cr1^gfp/+^ mice (Cx3cr1^gfp/gfp^ 697.89 µV ± 86.2 vs. Cx3cr1^gfp/+^ 542.58 µV ± 108.3, *n* = 9, Mann-Whitney-U, *p* = 0.007, Appendix A).

The oscillatory potentials at 2.5 cds/m^2^ showed a tendency to decrease 24 weeks after proton therapy in the heterozygous Cx3cr1^gfp/+^ mice and were mainly stable (Figure 4G). In direct comparison to the Cx3cr1^gfp/gfp^ the oscillatory potentials were significantly higher in the homozygous Cx3cr1^gfp/gfp^ mice (Cx3cr1^gfp/gfp^ irradiated 409.67 µV ± 59.9 vs. Cx3cr1^gfp/+^ irradiated 362.42 µV ± 21.5, *n* = 9/12, Mann-Whitney-U, *p* = 0.018, Appendix A). There was no significant difference between the irradiated and non-irradiated eyes.

The photopic wave amplitudes at 20cd (background 50 cd/m^2^) were stable for both genotypes up to 16 weeks after proton irradiation, but after 24 weeks they were significantly reduced in the homozygous Cx3cr1^gfp/gfp^ compared to the heterozygous Cx3cr1^gfp/+^ mice (*p* = 0.058, Figure 4F and Appendix A). There was no significant difference between the irradiated and non-irradiated eyes.

Overall, the ERG data show that the retinal function of the irradiated eyes of the homozygous Cx3cr1^gfp/gfp^ mice was less impaired compared to the other genotypes 24 weeks after proton irradiation. This mainly applies to the bipolar cell activity, the function of the pigment epithelium and of the amacrine cells as well as their interconnections.

In this regard, the lack of Cx3cr1 signaling seems to have a protective effect.

### 3.5. Cataract Formation in Cx3cr1 Knock out Animals

We next investigated whether the loss of Cx3cr1 had an impact on morphological parameters as well.

In all heterozygous Cx3cr1^gfp/+^ (*n* = 6) as well as in all homozygous Cx3cr1^gfp/gfp^ (*n* = 5) mice, 24 weeks after proton irradiation, imaging was not feasible due to cataract formation. An examination of the respective fellow eyes revealed no abnormalities 24 weeks after proton irradiation (Figure 5A).

Up to and including week 16 after proton irradiation, the retinal images of the irradiated eyes showed no significant changes in the autofluorescence compared to the non-irradiated control eye (Figure 5A).

Accordingly, there was also no difference in autofluorescence between the heterozygous Cx3cr1^gfp/+^ and the homozygous Cx3cr1^gfp/gfp^ mice (Figure 5A).

### 3.6. Inflammation in the Wild Type and Cx3cr1 Knock out Animals

To examine the cellular inflammatory component, we performed a FACS (fluorescence activated cell sort) analysis of CD11bCD45 positive cells of the retina. Both the proportion of microglia (CD11b^+^CD45^lo^) and monocytes (CD11b^+^CD45^hi^) were not significantly changed in the retina of the irradiated eye compared to the non-irradiated contralateral side (Appendix A). However, compared to the results in wild-type mice, the proportion of microglia (CD11bCD45^lo^) and that of monocytes (CD11bCD45^hi^) in the retinas of the heterozygous Cx3cr1^gfp/+^ and the homozygous Cx3cr1^gfp/gfp^ mice is significantly reduced (2-way-ANOVA, *n* = 8WT, 7Cx3cr1^gfp/+^, 10Cx3cr1^gfp/gfp^, *p* < 0.0001, Appendix A).

Using FACS, we cannot analyze the microglia population in different vascular layers of the retina. Therefore, we performed an immunofluorescence staining of Iba1 and Isolectin B4 in retinal flat mounts (Figure 5B).

In the irradiated retina of the WT 15 CGE- group, a noticeable accumulation of isolectin B4- positive cells was observed in the deep vascular layer (Figure 2A). These Isolectin B4 positive cells were not detected in the retinal flat mounts of the non-irradiated eyes, nor the radiated eyes of the 5 CGE- group or 10 CGE- group. Since isolectin B4 is also used to label microglia [30], we wanted to investigate the distribution of these cells using Iba1 as a co-staining more in depth.

There was a significant increase in the number of Iba1 positive cells per cubic millimeter in the retina of the irradiated eyes compared to that of the non-irradiated fellow eye in the wild-type mice (irradiated 9084.41 ± 2811.06 vs. non-irradiated 4359.63 ± 1689.01, *n* = 8, paired two-sample *t*-test, *p* < 0.0001) and both heterozygous Cx3cr1^gfp/+^ and the homozygous Cx3cr1^gfp/gfp^ mice (Cx3cr1^gfp/+^ irradiated 15,773.62 ± 4578.83 vs. non-irradiated 9277.18 ± 1172.68, *n* = 3, paired two-sample *t*-test, *p* < 0.0001; Cx3cr1^gfp/gfp^ irradiated 15,855.55 ± 2414.03 vs. non-irradiated 10,156.71 ± 1037.32, *n* = 3, paired two-sample *t*-test, *p* < 0.0001, Figure 5C). There was no significant difference between the Cx3cr1 genotypes. However, In both Cx3cr1 genotypes we detected an about two-fold higher number of green fluorescent cells than in the wild-type (Cx3cr1^gfp/+^ irradiated 15,773.62 ± 4578.83 vs. wild-type irradiated 9084.41 ± 2811.06, paired two-sample *t*-test, *p* < 0.0001; Cx3cr1^gfp/+^ non-irradiated 9277.18 ± 1172.68 vs. wild-type non-irradiated 4359.63 ± 1689.01, paired two-sample *t*-test, *p* < 0.0001; Cx3cr1^gfp/gfp^ irradiated 15,855.55 ± 2414.03 vs. wild-type irradiated 9084.41 ± 2811.06, paired two-sample *t*-test, *p* < 0.0001; Cx3cr1^gfp/gfp^ non-irradiated 10,156.71 ± 1037.32 vs. wild-type non-irradiated 4359.63 ± 1689.01, paired two-sample *t*-test, *p* < 0.0001)**.** Since in the Cx3cr1 genotypes microglia cells were labeled by EGFP and Iba1 cells were labeled using an Alexa-488-conjugated secondary antibody, the cells counted using the Alexa-488/EGFP filter are both, the Iba1 positive microglia cells and the EGFP positive microglial cells. This could be the reason why the number of Alexa-488/EGFP positive cells in the retina of the Cx3cr1 genotypes mice is increased compared to the wild-type mice. Therefore, a direct comparison between WT (only Iba1 positive microglia cells) and Cx3cr1 genotypes (Iba1 positive microglia cells and EGFP positive microglia cells) is not biological meaningful. We therefore focus on the comparison between irradiated and non-irradiated eyes within the respective genotypes.

Upon further investigation of the localization of the accumulation of Iba1 positive cells in the retina, concerning all three genotypes no significant difference was found regarding the cell number per cubic millimeter in the superficial vascular layer in comparison to the deep vascular layer (Appendix A) of the irradiated retinas.

In the non-irradiated control eyes of both the heterozygous and homozygous Cx3cr1 genotypes, however, it is noticeable that the number of retinal Iba1 positive microglia cells and EGFP positive microglia cells in the deep vascular layer is significantly higher than in the superficial vascular layer (Cx3cr1^gfp/+^ non-irradiated superficial 7903.67 ± 908.84 vs. deep vascular layer 10,650.69 ± 1764.64, paired two-sample *t*-test, *p* < 0.0001; Cx3cr1^gfp/gfp^ non-irradiated superficial 8988.02 ± 1408.29 vs. deep vascular layer 11,325.39 ± 1620.95, paired two-sample *t*-test, *p* = 0.004). In case of the irradiated eyes the opposite tendency can be observed. Here, the accumulation of Iba1 positive microglia cells and EGFP positive microglia cells seems to be greater in the superficial vascular layer greater (Cx3cr1^gfp/+^ irradiated superficial 16,829.05 ± 4717.62 vs. deep vascular layer 14,718.19 ± 5191.37, paired two-sample *t*-test, *p* > 0.05; Cx3cr1^gfp/gfp^ irradiated superficial 16,265.19 ± 2990.13 vs. deep vascular layer 15,445.91 ± 2458.11, paired two-sample *t*-test, *p* > 0.05). Thus, in the case of the Cx3cr1 genotype, because of irradiation, there is a more pronounced increase in the accumulation of Iba1 positive microglia cells and EGFP positive microglia cells in the superficial vascular layer (Appendix A).

### 3.7. Vascular Component and Gliosis/Fibrosis in Cx3cr1 Animals

Regarding the vascular component, the same result was observed for both Cx3cr1 genotypes as in the wild-type mice (Figure 6).

The number of NG2 positive cells per mm of vessel length was significantly reduced in the retina of the irradiated eye compared to the opposite eye (Cx3cr1^gfp/+^ irradiated 4.24 ± 0.8 vs. non-irradiated 8.3 ± 0.8, *n* = 5, Wilcoxon test, *p* = 0.043; Cx3cr1^gfp/gfp^ irradiated 4.39 ± 0.6 vs. non-irradiated 8.5 ± 1.3, *n* = 3, paired two-sample *t*-test, *p* = 0.015, Figure 6B). There was no difference between the images of the central retina and those of the retinal periphery (Appendix A).

In addition, there was also no difference in the number of NG2 positive cells per mm of vessel length in the superficial compared to the deep vascular layer (Appendix A).

Finally, there was no significant difference detectable between the three genotypes (Figure 6, Appendix A).

To investigate the influence of the Cx3cr1 signaling on gliosis and fibrosis, an immunofluorescence staining of GFAP and Vimentin was carried out in retinal sections 24 weeks after irradiation with 15 CGE (Figure 7).

Similar to the findings in the wild-type mice, the mean relative integrated density of GFAP was significantly increased on the images of the side of the irradiation compared to the opposite side (Cx3cr1^gfp/+^ irradiated 35.39 ± 2 vs. non-irradiated 33.39 ± 3, *n* = 4, paired two-sample *t*-test, *p* = 0.016; Cx3cr1^gfp/gfp^ irradiated 40.42 ± 6.7 vs. non-irradiated 33.48 ± 4.3, *n* = 4, paired two-sample *t*-test, *p* < 0.001, Figure 7B).

However, there was no significant difference in mean relative integrated density of GFAP between the genotypes or the central and peripheral areas (Figure 7C).

Regarding the vimentin staining, as with the wild-type mice, there was no significant difference between the irradiated and control eyes following the irradiation. The genotype also had no influence on the result. There was no significant difference between the Cx3cr1^gfp/gfp^ and Cx3cr1^gfp/+^ genotype (Figure 7).

## 4. Discussion

Proton irradiation has become the therapy of choice for central choroidal melanoma in humans, primarily due to its ability to deliver a highly focused radiation dose while minimizing damage to surrounding tissues [1,31,32,33]. However, radiation retinopathy remains a significant long-term complication, with a reported 5-year incidence rate of 85% [6,7]. Like diabetic retinopathy, it is characterized by a microangiopathy, including endothelial cell loss, ischemia, bleeding, and macular edema, leading to significant visual impairment [1,9,10].

Previous research has primarily focused on tumor control, with limited exploration of the pathophysiological mechanisms underlying radiation retinopathy, particularly in experimental models [1]. Our study is the first to use a targeted proton irradiation model for mouse eyes, enabling an investigation of dose-dependent vascular and retinal changes.

We observed a dose-dependent loss of pericytes in irradiated retinal vessels, resembling early diabetic retinopathy where pericyte loss is a hallmark [34]. Additionally, retinal gliosis, indicated by increased GFAP expression, was demonstrated in a dose-dependent manner, consistent with reactive glial activation reported in other retinal diseases [35].

Inflammatory responses were evidenced by increased Iba1-positive microglial cells post-irradiation. Interestingly, Cx3cr1 deficiency was associated with a functional protective effect, potentially due to restricted microglial infiltration into the deep vascular layer. This finding contrasts with diabetic retinopathy models, where Cx3cr1 deficiency exacerbated inflammation and vascular damage [36].

Functionally, retinal impairment was primarily observed in photoreceptor and bipolar cells, with significant alterations in the ERG a-wave and b-wave amplitudes at higher radiation doses. Despite the dense cataract development hindering in vivo observations at later stages, our findings provide insights into the structural and functional impact of proton irradiation on the retina and highlight the need for further mechanistic studies on the role of the Cx3cr1 pathway in mitigating radiation-induced damage.

In humans, the presence of radiation retinopathy is primarily assessed by performing fluorescence angiography of the retinal vessels. The characteristic findings are severe retinal capillary nonperfusion, capillary dilation, and microaneurysms, frequently in combination with macular edema or ischemia [9].

Therefore, retinal fluorescence angiography was performed in the mice in week 8, 16 and 24 after proton therapy. However, no gross vessel abnormalities could be detected. The main side effect of irradiation was a dose and time-dependent development of dense cataracts. Therefore an in vivo examination of eyes at 24 weeks after irradiation with 15 CGE was impossible. Hence, at 24 weeks, changes in the sense of a radiation retinopathy could not be detected in vivo, but also cannot be excluded, at least for the 15 CGE group. Thus, in vivo, we were able to perform FAG and subsequent vessel analysis only prior to cataract formation. While a few samples were collected despite cataract formation and prepared as flat mounts, vessel abnormalities were observed. However, our focus was on investigating changes before cataract formation to align these findings with the in vivo FAG data. We acknowledge that, in a follow-up study, it would be valuable to examine eyes that developed cataracts, but this was not the primary objective of the current study.

A key finding of previous publications on the pathophysiology of radiation retinopathy is a microangiopathy that seems to be similar to diabetic retinopathy [9].

The loss of endothelial cells and associated pericytes have already been described as primary vascular event in experiments with other forms of ionizing radiation and other species such as rats [10,37].

In the context of this experimental study, we showed for the first time a dose-dependent loss of NG2 positive pericytes surrounding Isolectin B4 positive endothelial cells of retinal vessels of the irradiated mouse eyes. This dose-dependent loss of pericytes seems to precede an endothelial loss, showing a similarity to early diabetic retinopathy, where pericyte loss is considered as main feature [34]. This stands in contrast to the publication by Gardiner et al. [11], who reported that in the context of radiation retinopathy, there is a loss of microvascular endothelial cells in the presence of surviving pericytes, as demonstrated through vascular digests obtained from both clinical radiation retinopathy and animal models. In this experimental study, Isolectin B4 was utilized to label endothelial cells for the visualization and analysis of vascular architecture, though it does not provide direct information regarding cell viability. Consequently, we can only conclude that, in comparison to the length of Isolectin B4-positive vessels, there is a dose-dependent loss of pericytes. However, we cannot ascertain the impact of radiation on the viability of endothelial cells based on our data. Additionally, the experimental designs differ; we used proton radiation as opposed to photon radiation, utilized a different animal model, and applied different radiation intensities. It would be valuable to further investigate how these differing irradiation modalities might influence the results regarding pericyte and endothelial cell interaction.

Possible degenerative processes were examined by the means of an immunofluorescence staining of GFAP and vimentin to show activated retinal glial cells as a non-specific neuropathological reaction. An increasing astrocyte and Mueller cell reactive gliosis, evident by upregulation of GFAP expression, could also be demonstrated in a dose-dependent manner. No increased expression of GFAP was reported in the central retina 3 months after irradiation in the context of a mouse model of chronic glaucoma [35]. However, since the type and dose of irradiation as well as the time frame differ from our experimental setting, a direct comparison is not possible.

Since the severity of the phenotype was more evident in the 15 CGE group, the further experiments were carried out with this radiation dose. And since the results of the non-irradiated eyes correspond to those of the sham eyes, the sham group was dispensed in the subsequent experiments.

While in X-radiation (2–20 Gy) an detrimental effect on photoreceptors was reported [38], recent publications indicate that photoreceptors, as non-replicating cells, are relatively stable to damage from radiation (e.g., brachytherapy, external beam radiotherapy, proton beam radiation, helium ion radiotherapy, and gamma knife radiotherapy at approximately 45 Gy) [39]. However, impairment of the retinal pigment epithelium through radiation has already been described. It is characterized by a loss of melanin, an accumulation of lipofuscin and changes in choroidal vessels [14,39]. There are only a few publications on ERG and radiation retinopathy. Fujii et al. described in a book chapter that there was no significant change in the ERG results after beta radiation with 51 Gy of rabbit eyes [40].

As part of this publication, the retinal function after proton irradiation, including the photoreceptor activity and function of the retinal pigment epithelium, was examined using the Ganzfeld ERG. In wild-type mice, no significant difference between the irradiated and control eyes could be observed. However, there was a tendency towards impairment of the retinal function after proton irradiation, especially of the photoreceptor activity and bipolar cells.

The role of inflammatory processes in the pathogenesis of radiation retinopathy is unclear. Clinical data and the success of treatment with corticosteroids suggest that an inflammatory reaction may be involved. In the few existing publications, microglia seem to play a major role [1,41,42].

The results from the FACS experiments demonstrate no significant difference between irradiated and control eyes regarding the proportion of microglia (CD11b^+^CD45^lo^) and monocytes (CD11b^+^CD45^hi^) in the entire retina. These surface markers therefore do not seem to be increasingly expressed following proton radiation.

In contrast, the immunofluorescence staining of Iba1 in retinal flat mounts of the wild- type showed a significantly accumulation of Iba1 positive cells in the retina on the side of the irradiation compared to the control eye. Subsequently, the experiments were repeated with heterozygous Cx3cr1^gfp/+^ and homozygous Cx3cr1^gfp/gfp^ mice to investigate the extent of inflammatory involvement respectively the influence of the Cx3cr1 signaling pathway on the processes in the retina after proton irradiation.

The Cx3cr1/CX3CL1 signaling pathway is mainly used to recruit monocytes to regions with injury and inflammation [43]. This also applies to the retina. In the case of a disturbed blood-retina barrier, for example, recruited monocytes can be detected in the retina [44]. Otherwise, in the retina the Cx3cr1 receptor is constant, invariably and apparently exclusively expressed by retinal microglial cells [18]. It has already been published that Cx3cr1 has an important role in trafficking microglia cells to and from the subretinal space and that the ligand CX3CL1 displays pro-angiogenic activity [18].

In general, the increased expression of Iba1 in microglia and macrophages correlates with increased inflammatory activity [28]. Most recently, the increase in Iba-1 was associated with a detrimental microglia phenotype, called “dark microglia” [45]. In the present publication, it could be demonstrated for the first time that targeted proton irradiation of eyes leads to a significant inflammatory reaction within the retina, maybe pushing the microglia towards the “dark” side.

In the literature, so far, no influence of the Cx3cr1 signaling pathway on the number of retinal pericytes has been described.

Comparable to wild-type mice, there is a significant decrease in the number of pericytes in the retinal flat mounts of the irradiated eyes of both Cx3cr1 genotypes, evenly in all areas and layers. Based on our experiments, the CX3cr1 signaling does not seem to have any influence on the decrease in the number of pericytes because of the irradiation.

According to the findings with the wild-type mice, the proton irradiation of both Cx3cr1 genotype mice also resulted in a significant increase in the expression of GFAP in the retina. However, there is no significant difference between the genotypes and no significant difference between the central and peripheral areas of the irradiated retina.

In our experiments, neither the proton irradiation in general nor the genotype respectively the absence of the Cx3cr1 signal, did affect the expression of Vimentin as a structural marker of Müller cells in the retina.

Regarding retinal function, ERG results from the irradiated eyes of heterozygous Cx3cr1^gfp/+^ and homozygous Cx3cr1^gfp/gfp^ mice tend to be consistently reduced than the results of the control eyes 24 weeks after proton irradiation, which correspond to the results of the wild-type mice. Photoreceptor and bipolar cell activity are mainly affected.

In homozygous Cx3cr1^gfp/gfp^ mice, a-waves and b-wave amplitudes at 3 cds/m^2^ were significantly worse on the irradiated side in relation to the contralateral side. However, the retinal function of homozygous Cx3cr1^gfp/gfp^ mice seems to be generally better than of the heterozygous Cx3cr1^gfp/+^ and wildtype mice 24 weeks after irradiation. This difference is even significant regarding the function of the pigment epithelium and of the amacrine cells as well as their interconnections. The absence of the Cx3cr1 signaling obviously has a positive effect on the loss of retinal function after proton irradiation.

Like the findings in wild type, the density of Iba1/Cx3cr1 positive cells is increased after irradiation in both Cx3cr1 genotypes. In contrast to the results of the wild-type genotype, the distribution of Iba1/Cx3cr1 positive cells is different. In the wild-type genotype we observe an even increase in the superficial and the deep vascular layer, whereas in both Cx3cr1 genotypes the increase is more in the superficial than in the deep vascular layer. Since Cx3cr1 is related to trafficking of these cells, the cells might get “stuck” in the superficial layer and do not migrate to the deep vascular layer. Since the loss of Cx3cr1 seems to be protective on a functional level as measured with the ERG, the penetration of Iba1/Cx3cr1 positive cells into the deep vascular layer might be a hallmark of radiation-induced retinopathy.

In experimental diabetic retinopathy, there is also an increase in the number of activated microglial cells and migration of these cells into the outer plexiform layer and the photoreceptor layer [46]. In this case Cx3cr1 deficiency activated microglia, enhanced the inflammatory response, disrupted the vascular integrity and accelerated the progression of diabetic retinopathy [36].

In line with our findings, one other recent study using 45 Gy photon irradiation has shown increased inflammation and subsequent cell loss due to enhanced microglia activation and migration throughout the whole retina [17].

In the context of this publication, the experimentally proton irradiation of mouse eyes was carried out for the first time and the consequences of this ocular irradiation were investigated. The morphological findings after irradiation are essentially characterized by the development of a dense cataract. A degenerative component is reflected by an increased expression of GFAP in the retina and the increased accumulation of Iba1 positive cells demonstrates an involvement of inflammatory processes. In addition, an impairment of retinal function can be observed. The Cx3cr1 signaling pathway appears to have at least a positive effect on the retinal function that might be due to a prevention of the infiltration of Cx3cr1/Iba positive cells into the deep vascular layer. Overall, the model is valid to study radiation-related changes in vivo. However, the impact of the Cx3cr1 and its mechanistic role regarding microglia migration and function is still speculative and should be investigated further. Functionally, while photoreceptor and bipolar cell activity were primarily affected, the general preservation of retinal function in Cx3cr1-deficient mice offers a promising avenue for therapeutic intervention. The protective role of Cx3cr1 deficiency highlights the need to explore strategies that selectively modulate microglial activity and migration without compromising their physiological roles in retinal homeostasis.

Given the limited therapeutic options currently available, these findings provide a foundation for developing targeted strategies aimed at preserving vascular integrity, modulating glial and inflammatory responses, and ultimately improving outcomes for patients undergoing proton irradiation. Future research should focus on the mechanistic role of the Cx3cr1 pathway and explore pharmacological interventions that can be translated into clinical settings such as intravitreal Injections of antibodies against Cx3cr1.

## Figures and Tables

**Figure 1 cells-14-00298-f001:**
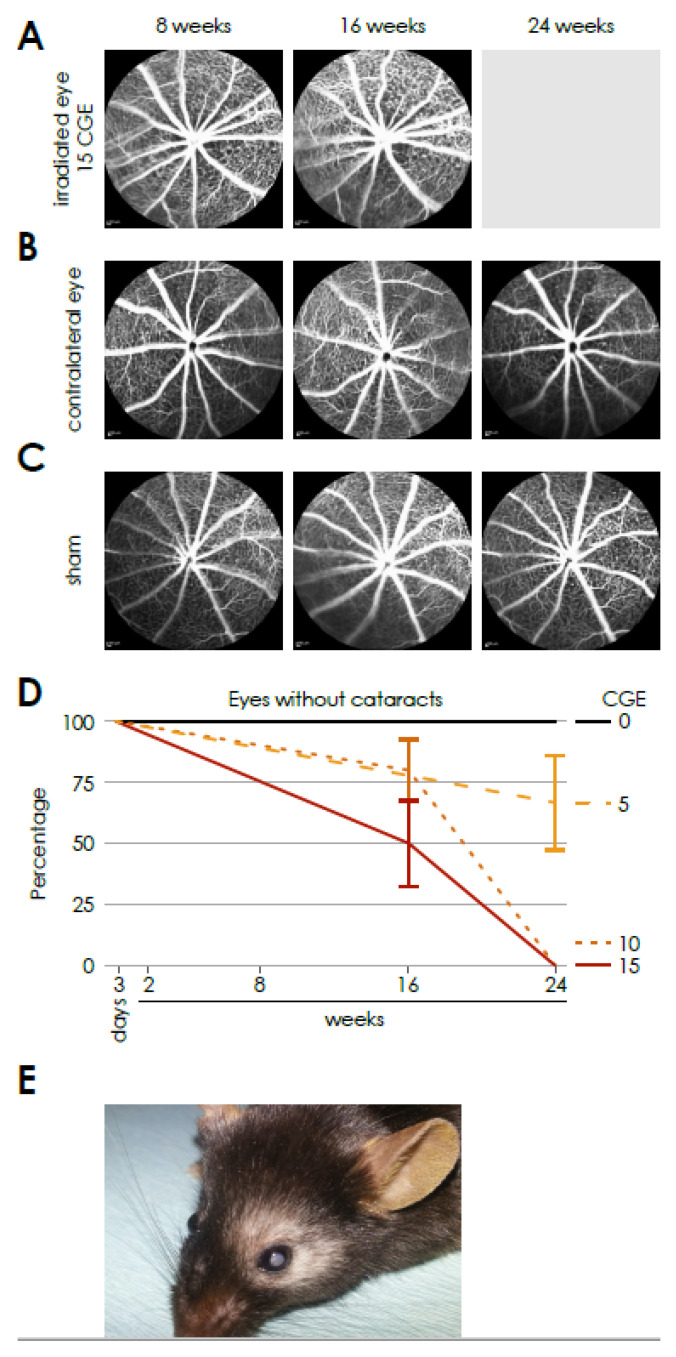
Vascular dynamics in the retina of the wild-type genotype after proton irradiation, time-course of fundus angiographies after proton irradiation in the irradiated eye (**A**), in the contralateral eye (**B**) and after sham procedure (**C**) as well as the development of cataract (**D**) with an exemplary photo (**E**).

**Figure 2 cells-14-00298-f002:**
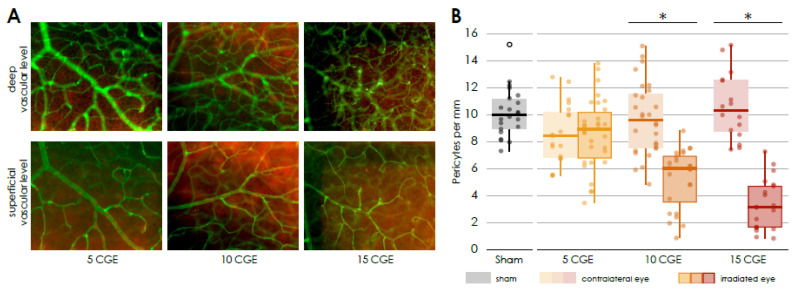
Reduced number of pericytes in the irradiated retina (**A**) Immunofluorescence staining of pericytes in the central retinal flat mounts, Isolectin B4 (FITC) co NG2 (Cy3), of irradiated eyes (with 5, 10 and 15 CGE), upper row: deep vascular layer, * activated microglia, lower row superficial vascular layer, pericytes (white arrowhead). (**B**) Number of pericytes per mm vessel length in the retina, significantly decreased number of pericytes per mm in the irradiated eye in relation to the contralateral side and sham group for the doses 10 CGE (*p* < 0.001) and 15 CGE (*p* = 0.002).

**Figure 3 cells-14-00298-f003:**
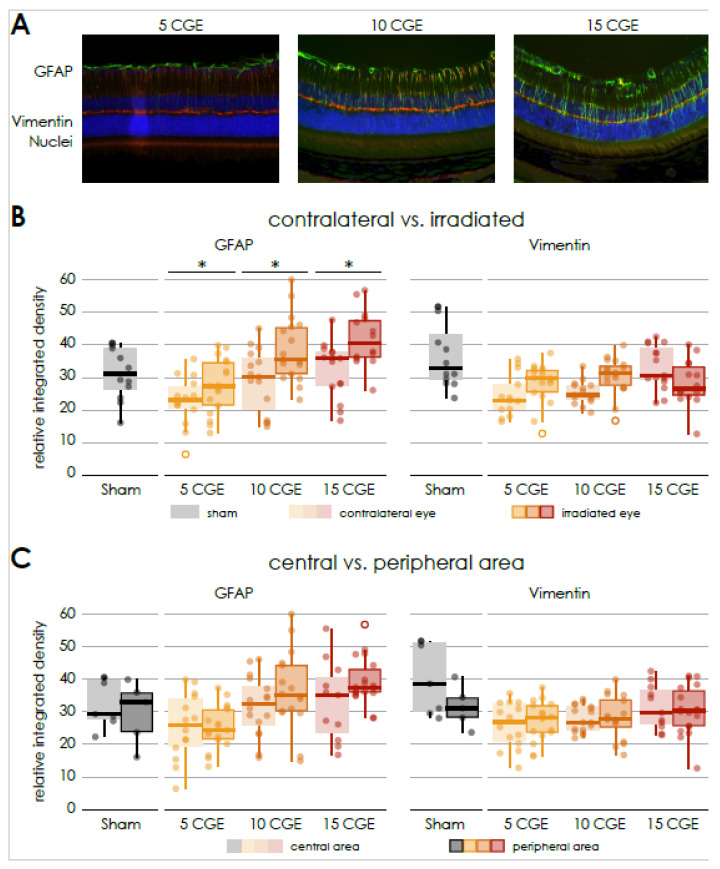
Gliosis in retinal sections of wild-type mice following irradiation (**A**) Immunofluorescence staining of central retinal sections of wild-type mice, GFAP (FITC) co Vimentin (Cy3), DAPI, of irradiated eyes (with 5, 10 and 15 CGE). (**B**) Relative integrated density of GFAP and Vimentin, irradiated versus contralateral side, GFAP: significant difference for every proton beam doses, no significance concerning sham (5 CGE: *p* < 0.001, 10 CGE: *p* = 0.007, 15 CGE: *p* < 0.001), Vimentin: no significant difference. (**C**) Relative integrated density of GFAP and Vimentin, central versus peripheral area, GFAP, Vimentin: no significant difference * *p* < 0.001.

**Figure 4 cells-14-00298-f004:**
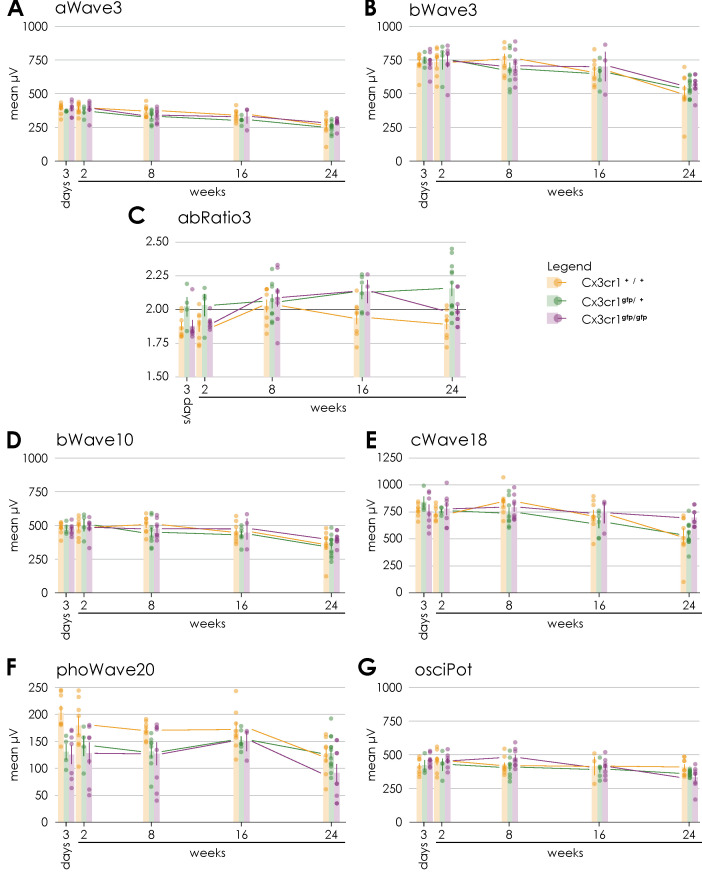
Comparison of Ganzfeld-ERG parameters regarding the genotype over time after proton irradiation with 15 CGE: (**A**) no significant difference between the genotypes, (**B**) no significant difference between the genotypes, (**C**) significant difference between the irradiated eyes of heterozygous Cx3cr1^gfp/+^ and wild-type mice (*p* = 0.003) or homozygous Cx3cr1^gfp/gfp^ (*p* = 0.049), (**D**) tendency difference between the irradiated eyes of homozygous Cx3cr1^gfp/gfp^ and heterozygous Cx3cr1^gfp/+^ mice (*p* = 0.072), (**E**) significant difference between the irradiated eyes of homozygous Cx3cr1^gfp/gfp^ and heterozygous Cx3cr1^gfp/+^ mice) (*p* = 0.007) or wild-type mice (*p* = 0.043), (**F**) tendency difference between the irradiated eyes of homozygous Cx3cr1^gfp/gfp^ and heterozygous Cx3cr1^gfp/+^ mice (*p* = 0.058), (**G**) significant difference between the irradiated eyes of homozygous Cx3cr1^gfp/gfp^ and heterozygous Cx3cr1^gfp/+^ mice (*p* = 0.018).

**Figure 5 cells-14-00298-f005:**
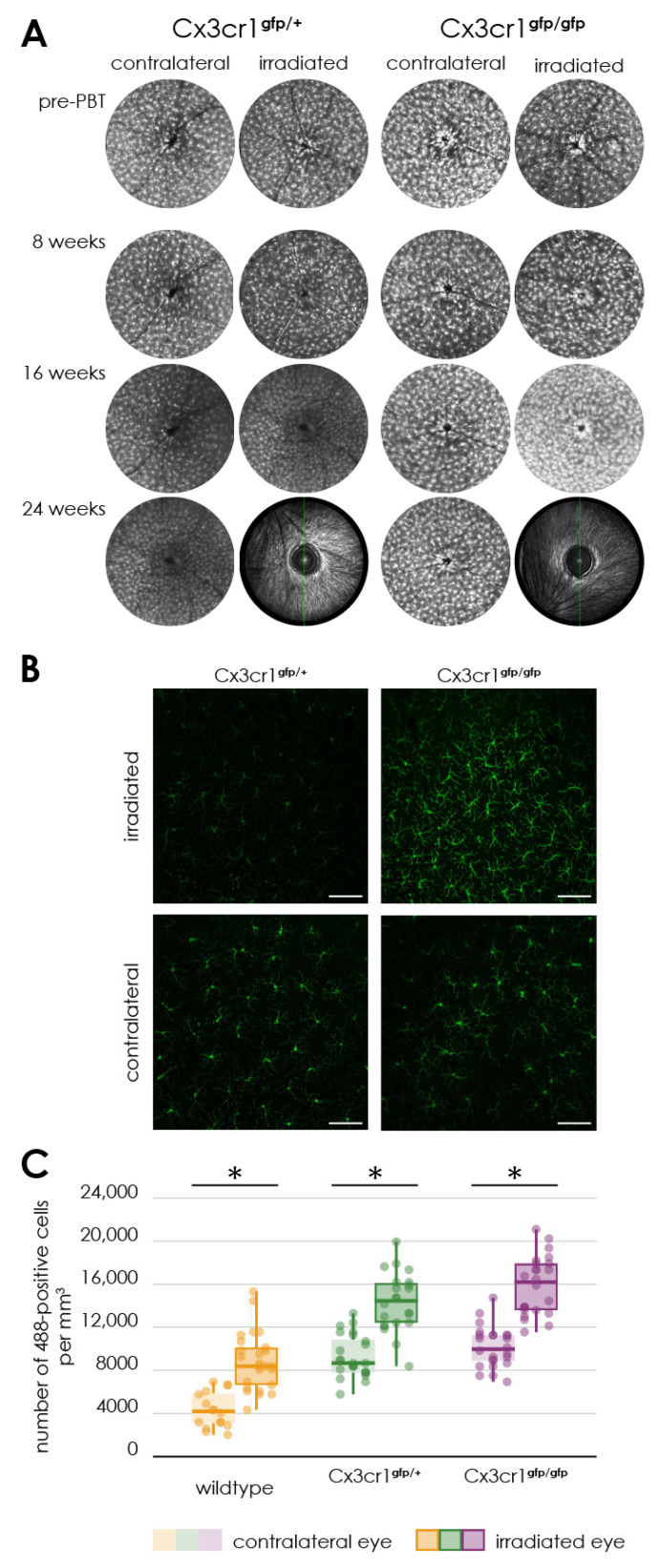
Activation of microglia following proton irradiation, (**A**) autofluorescence imaging of the retina of transgenic Cx3cr1^gfp/gfp^ and Cx3cr1^gfp/+^ mice after proton irradiation, time-course of fundus autofluorescence’s before and 8, 16 as well as 24 weeks after proton irradiation in the irradiated and contralateral eye of Cx3cr1^gfp/gfp^ and Cx3cr1^gfp/+^ mice, (**B**) immunofluorescence staining of Iba1 in retinal flat mounts of transgenic Cx3cr1^gfp/+^ and Cx3cr1^gfp/gfp^ mice after proton irradiation with 15 CGE, Scale bar: 86 µm, (**C**) number of Iba1-positive cells per cubic millimeter, comparison of the genotypes significantly increased number of Iba1-positive cells in the retina of irradiated eyes of the wildtype (* *p* < 0.001), Cx3cr1^gfp/gfp^ (* *p* < 0.001) and Cx3cr1^gfp/+^ mice (* *p* < 0.001) in relation to the contralateral side for the doses 15 CGE, no significant difference between the genotypes (*p* > 0.05).

**Figure 6 cells-14-00298-f006:**
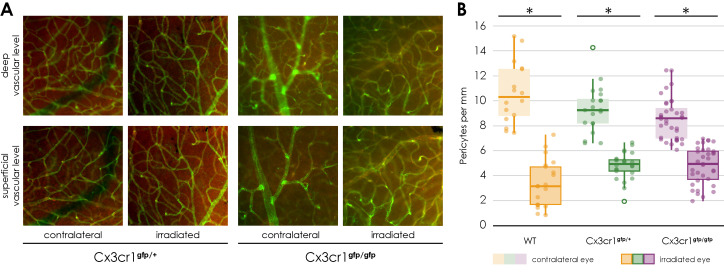
Immunofluorescence staining of NG 2 and Isolectin B4 in retinal flat mounts of transgenic Cx3cr1^gfp/+^ (a–d) and Cx3cr1^gfp/gfp^ mice after proton irradiation with 15 CGE, (**A**) upper row: deep vascular layer, lower row: superficial vascular layer (**B**) number of pericytes per mm vessel length 24 weeks after proton irradiation, comparison of the genotypes significantly decreased number of pericytes in the irradiated eye in relation to the contralateral side for the doses 15 CGE in Cx3cr1^gfp/gfp^ (* *p* = 0.015), Cx3cr1^gfp/+^ (* *p* = 0.043) and wild-type mice (* *p* = 0.002); no difference between genotypes, tendency: wild-type mice have more pericytes per mm in contralateral eye than transgenic Cx3cr1 mice.

**Figure 7 cells-14-00298-f007:**
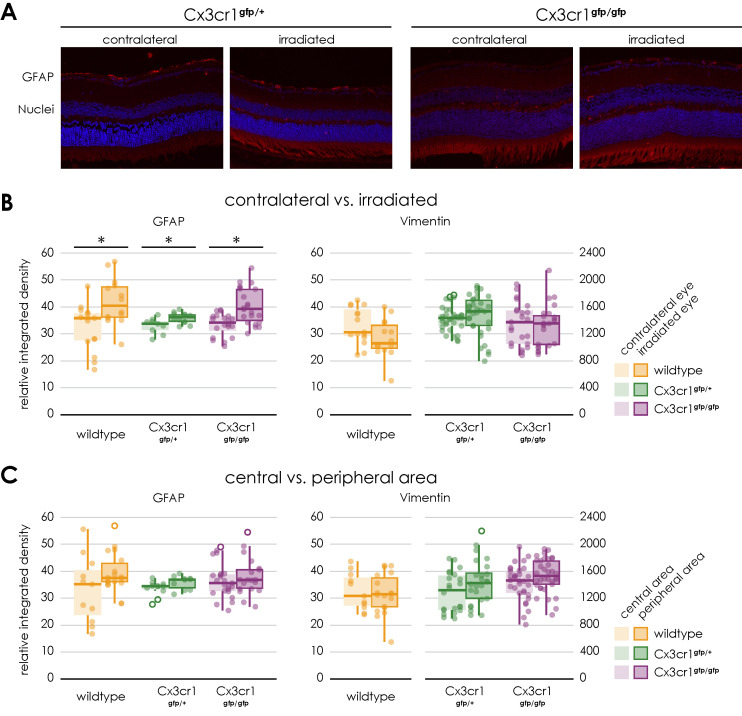
Gliosis in retinal sections of transgenic Cx3cr1^gfp/+^ and Cx3cr1^gfp/gfp^ mice following irradiation, (**A**) Immunofluorescence staining of GFAP in central retinal sections of transgenic Cx3cr1^gfp/+^ (left) and Cx3cr1^gfp/gfp^ mice (right) after proton irradiation with 15 CGE, contralateral non-irradiated versus irradiated retina, (**B**) relative integrated density of GFAP and Vimentin, all genotypes, irradiated versus contralateral side, GFAP: Increased integrated density with significant difference between the irradiated eye and the contralateral side for the Cx3cr1^gfp/+^ (* *p* = 0.016) and Cx3cr1^gfp/gfp^ genotype (* *p* < 0.001), no significant difference between the genotypes (*p* > 0.05), Vimentin: no significant difference (*p* > 0.05), (**C**) relative integrated density of GFAP and Vimentin, central versus peripheral area GFAP, Vimentin: no significant difference.

## Data Availability

The original contributions presented in this study are included in the article/Appendix A. Further inquiries can be directed to the corresponding author. The raw data supporting the conclusions of this article will be made available by the authors on request.

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
