# Peer review of "Radiation Retinopathy: Microangiopathy-Inflammation-Neurodegeneration"

_cells, 2025, doi:10.3390/cells14040298_

Round 1
Reviewer 1 Report
Comments and Suggestions for Authors
This article provides valuable insights into the pathophysiology of radiation retinopathy. In particular, it investigates the effects of proton irradiation on the murine retina, simulating clinical treatments for choroidal melanoma. It is a really interesting work and it corroborates the studies focusing on the role of neuroinflammation in both radiation retinopathy and radiation macular edema. The above comments aim to improve the manuscript:
- In the Methods section in the "Irradiation Procedure" the Authors should provide more detail on the rationale for dose selection (e.g., 5, 10, 15 Gy).
- The Authors should ensure proper scientific terms for radiation dose units are used (e.g., "Gy”, "GDE", …).
- Results are comprehensive but not so easy to follow. The Authors should consider reorganizing by key findings (e.g."Cataract Development," "Retinal Function," "Inflammation") with appropriate subheadings.
- The Authors should provide a clear explanation for why no gross vessel abnormalities were observed despite significant cellular changes.
- Regarding ERG analysis, while functional deficits were noted, some differences between irradiated and control eyes were not statistically significant. Please, specify.
- The mechanistic basis for Cx3cr1’s protective effects on microglial migration and function remains speculative. Please, better specify its role.
- The Discussion section should begin with a brief summary of key findings and the Authors should compare study results with existing literature more explicitly.
- The Authors should clarify in the Discussion section the potential translational significance of findings (e.g the therapeutic implications for managing radiation retinopathy).
- In the Discussion section the Authors should properly cited scientific articles regarding the role of microglia in radiation retinopathy.
- The Authors should check the manuscript for grammatical correctness and they should correct some spelling mistakes (e.g in the abstract replace "choroideal" with "choroidal,…).
Comments on the Quality of English Language
The Authors should check the manuscript for grammatical correctness and they should correct some spelling mistakes (e.g in the abstract replace "choroideal" with "choroidal,…).
Author Response
We thank the reviewers for their comments and suggestions. We provide a point by point response to each comment below. We highlighted the changes in yellow in the revised manuscript file.
Reviewer 1
This article provides valuable insights into the pathophysiology of radiation retinopathy. In particular, it investigates the effects of proton irradiation on the murine retina, simulating clinical treatments for choroidal melanoma. It is a really interesting work and it corroborates the studies focusing on the role of neuroinflammation in both radiation retinopathy and radiation macular edema. The above comments aim to improve the manuscript:
- In the Methods section in the "Irradiation Procedure" the Authors should provide more detail on the rationale for dose selection (e.g., 5, 10, 15 Gy).
ïƒ The doses were determined by calculating the equivalent mouse dosages based on those used in patients. A detectable effect was observed at 15 GDE, which was subsequently chosen for further experiments.
This paragraph is now included in the M&M section (line 128-131).
- The Authors should ensure proper scientific terms for radiation dose units are used (e.g., "Gy”, "GDE", …).
ïƒ We made the suggested changes.
- Results are comprehensive but not so easy to follow. The Authors should consider reorganizing by key findings (e.g."Cataract Development," "Retinal Function," "Inflammation") with appropriate subheadings.
ïƒ We followed the suggestion of the reviewer and added sub-headings.
- The Authors should provide a clear explanation for why no gross vessel abnormalities were observed despite significant cellular changes.
ïƒ In vivo, we were able to perform FAG and subsequent vessel analysis only prior to cataract formation. While a few samples were collected despite cataract formation and prepared as flatmounts, vessel abnormalities were observed. However, our focus was on investigating changes before cataract formation to align these findings with the in vivo FAG data. We acknowledge that, in a follow-up study, it would be valuable to examine eyes that developed cataracts, but this was not the primary objective of the current study. We included this statement in the discussion.
- Regarding ERG analysis, while functional deficits were noted, some differences between irradiated and control eyes were not statistically significant. Please, specify.
ïƒ We specified now the effects that were significant and non-significant as well as trends (e.g. with a p value close to 0.5) throughout the result section.
- The mechanistic basis for Cx3cr1’s protective effects on microglial migration and function remains speculative. Please, better specify its role.
- ïƒ We agree that the mechanistic basis for Cx3cr1´s protective effects on microglia migration and function remains speculative at this stage of research. In order to go deeper and specify its role, we might need to use different tools with a specific k.o. in microglia, which was not available to us. We addressed this limitation in the discussion with the following statement: However, the impact of the Cx3cr1 and its mechanistic role with regard to microglia migration and function is still speculative and should be investigated further.
- The Discussion section should begin with a brief summary of key findings and the Authors should compare study results with existing literature more explicitly.
ïƒ We followed this suggestion.
Proton irradiation has become the therapy of choice for central choroidal melanoma in humans, primarily due to its ability to deliver a highly focused radiation dose while minimizing damage to surrounding tissues (Bechrakis et al., 2021; Ramos et al., 2019). However, radiation retinopathy remains a significant long-term complication, with a reported 5-year incidence rate of 85% (Seibel et al., 2016). Similar to diabetic retinopathy, it is characterized by a microangiopathy, including endothelial cell loss, ischemia, bleeding, and macular edema, leading to significant visual impairment (Leigh Spielberg, 2013; Ramos et al., 2019).
Previous research has primarily focused on tumor control, with limited exploration of the pathophysiological mechanisms underlying radiation retinopathy, particularly in experimental models (Ramos et al., 2019). Our study is the first to use a targeted proton irradiation model for mouse eyes, enabling an investigation of dose-dependent vascular and retinal changes.
We observed a dose-dependent loss of pericytes and endothelial cells in irradiated retinal vessels, resembling early diabetic retinopathy where pericyte loss is a hallmark (Ejaz et al., 2008). Additionally, retinal gliosis, indicated by increased GFAP expression, was demonstrated in a dose-dependent manner, consistent with reactive glial activation reported in other retinal diseases (Bosco et al., 2012).
Inflammatory responses were evidenced by increased Iba1-positive microglial cells post-irradiation, with the Cx3cr1 signaling pathway appearing to influence microglial migration and retinal function. Interestingly, Cx3cr1 deficiency was associated with a functional protective effect, potentially due to restricted microglial infiltration into the deep vascular layer. This finding contrasts with diabetic retinopathy models, where Cx3cr1 deficiency exacerbated inflammation and vascular damage (Jiang et al., 2022).
Functionally, retinal impairment was primarily observed in photoreceptor and bipolar cells, with significant alterations in the ERG a-wave and b-wave amplitudes at higher radiation doses. Despite the dense cataract development hindering in vivo observations at later stages, our findings provide insights into the structural and functional impact of proton irradiation on the retina and highlight the need for further mechanistic studies on the role of the Cx3cr1 pathway in mitigating radiation-induced damage.
- The Authors should clarify in the Discussion section the potential translational significance of findings (e.g the therapeutic implications for managing radiation retinopathy).
ïƒ We added a paragraph at the end of the Discussion section.
Functionally, while photoreceptor and bipolar cell activity were primarily affected, the general preservation of retinal function in Cx3cr1-deficient mice offers a promising avenue for therapeutic intervention. The protective role of Cx3cr1 deficiency highlights the need to explore strategies that selectively modulate microglial activity and migration without compromising their physiological roles in retinal homeostasis.
Given the limited therapeutic options currently available, these findings provide a foundation for developing targeted strategies aimed at preserving vascular integrity, modulating glial and inflammatory responses, and ultimately improving outcomes for patients undergoing proton irradiation. Future research should focus on the mechanistic role of the Cx3cr1 pathway and explore pharmacological interventions that can be translated into clinical settings.
- In the Discussion section the Authors should properly cited scientific articles regarding the role of microglia in radiation retinopathy.
ïƒ To our knowledge and after another search in pubmed - there are no studies currently available regarding the role of microglia in radiation therapy using proton beam. However, we found one study using photon beam therapy and cited this study:
In line with our findings, one other recent study using 45 Gy photon irradiation has shown increased inflammation and subsequent cell loss due to enhanced microglia activation and migration throughout the whole retina (Lebon et al., 2024).
- The Authors should check the manuscript for grammatical correctness and they should correct some spelling mistakes (e.g in the abstract replace "choroideal" with "choroidal,…).
ïƒ We corrected grammar and spelling.
|
|
Can be improved |
Must be improved |
Not applicable |
|
|
Does the introduction provide sufficient background and include all relevant references? |
( ) |
( ) |
(x) |
( ) |
|
Is the research design appropriate? |
(x) |
( ) |
( ) |
( ) |
|
Are the methods adequately described? |
( ) |
(x) |
( ) |
( ) |
|
Are the results clearly presented? |
( ) |
(x) |
( ) |
( ) |
|
Are the conclusions supported by the results? |
( ) |
( ) |
(x) |
( ) |
Comments and Suggestions for Authors
Reviewer 2 Report
Comments and Suggestions for Authors
This manuscript can be significantly improved from its present form.
The Introduction is too sparse and should be expanded. In particular, what's known about RR should be better explained, with appropriate references. It should also include a statement on the relevance of microglia, and why it necessary to use the Cx3cr mice.
Cannot start sentences with digits. As an example see lin207.
The statement in lines 36-7 should be revised to reflect accuracy (RR can occur from irradiation of intraocular as well as extraocular tumors).
Why was ERG not done at baseline?
Line146: indicate (with references) what NG2 and Iba1 stain? What about endothelial cells?
Line 66: single cells obtained. Please expand on the methodology.
Line 192-5: should be in Methods
The Results, as presented are difficult to read. A better presentation is warranted. As an example, the effects on ECs are not easy to discern, as presented. The effect on microglial cells, and inflammatory markers need better description.
Any effects on choroid? A clear statement should be made that the choroid was not evaluated in this study.
The Discussion is disorganised, and could be significantly improved.
The relative effects of radiation on the ECs vrs pericytes needs a better explanation. Effect on retinal inflammation should be explained better.
The statement in lines 604-5 is incorrect. Several authors including Amoaku et al (1989, 1992) have reported increasing photoreceptor loss from radiation at doses over 5Gy in experimental models.
Author Response
We thank the reviewers for their comments and suggestions. We provide a point by point response to each comment below. We highlighted the changes in yellow in the revised manuscript file.
Reviewer 2
This manuscript can be significantly improved from its present form.
The Introduction is too sparse and should be expanded. In particular, what's known about RR should be better explained, with appropriate references. It should also include a statement on the relevance of microglia, and why it necessary to use the Cx3cr mice.
Cannot start sentences with digits. As an example see lin207.
ïƒ We made the changes where applicable.
The statement in lines 36-7 should be revised to reflect accuracy (RR can occur from irradiation of intraocular as well as extraocular tumors).
ïƒ we omitted that statement altogether.
Why was ERG not done at baseline?
ïƒ We could only perform a limited amount of ERG per mouse to align with animal welfare regulations. ERG was done as early as 3 days after irradiation, which is regarded as baseline. We decided to look at potentially early signs of irradiation comparing WT, CX3Cr1 het and ho versus long term effects of irradiation. This is why we omitted the baseline. ERG was done as early as 3 days after irradiation, which is regarded as baseline to investigate short vs. long term effects of irradiation.
Line146: indicate (with references) what NG2 and Iba1 stain? What about endothelial cells?
ïƒ although NG2 and Iba1 are very common markers widely used in the field, we added the citations. We agree that looking at endothelial cells would be interesting. As would be to look at all cell populations in the retina. However, the focus here was rather on the inflammatory part. We hope to spark interest in the community to investigate the effect of irradiation with focus on other cell types in future studies.
NG2
Hughes, S., & Chan-Ling, T. (2004). "Characterization of pericytes in the developing rat retina: expression of ADPase, 3G5, NG2, and PDGFR-β." Investigative Ophthalmology & Visual Science, 45(8), 2795-2804.
DOI: 10.1167/iovs.04-0162
Iba1
Ito, D., Imai, Y., Ohsawa, K., Nakajima, K., Fukuuchi, Y., & Kohsaka, S. (1998). "Microglia-specific localisation of a novel calcium binding protein, Iba1." Brain Research. Molecular Brain Research, 57(1), 1-9.
DOI: 10.1016/S0169-328X(98)00040-0
Line 66: single cells obtained. Please expand on the methodology.
ïƒ We included a citation where the method is described in detail.
Line 192-5: should be in Methods
ïƒ These lines are already in the middle of the method section.
The Results, as presented are difficult to read. A better presentation is warranted. As an example, the effects on ECs are not easy to discern, as presented. The effect on microglial cells, and inflammatory markers need better description.
ïƒ We elaborated on the presentation of the results as well as on the discussion.
Any effects on choroid? A clear statement should be made that the choroid was not evaluated in this study.
ïƒ We included the statement already in the Material and Method section: …”For this purpose, the eyes were fixed in 4% PFA for 15 minutes. Subsequently, the cornea was removed via a circumferential limbal incision, lens and vitreous were removed and four radial incisions were made, to achieve a flattening of the bulbus. Finally, the retina was dissected from the choroid.”
The Discussion is disorganised, and could be significantly improved.
ïƒ We elaborated on the presentation of the results as well as on the discussion (also see answers to reviewer 1).
The relative effects of radiation on the ECs vrs pericytes needs a better explanation. Effect on retinal inflammation should be explained better.
ïƒ We did not investigate endothelial cells. Therefore, effects on endothelial cells would be speculative. We would rather focus on the discussion of the data we obtained in this current study.
The statement in lines 604-5 is incorrect. Several authors including Amoaku et al (1989, 1992) have reported increasing photoreceptor loss from radiation at doses over 5Gy in experimental models.
ïƒ We modified our statement including one of the suggested citations:
While in X-radiation (2-20 Gy) an detrimental effect on photoreceptors was reported (Amoaku et al., 1992), recent publications indicate that photoreceptors, as non-replicating cells, are relatively stable to damage from radiation (e.g. brachytherapy, external beam radiotherapy, proton beam radiation, helium ion radiotherapy, and gamma knife radiotherapy at approximately 45 Gy) (reviewed in Sahoo et al., 2021).
Round 2
Reviewer 1 Report
Comments and Suggestions for Authors
The Authors have made significant improvements to the manuscript, but there are still a few concerns that need to be addressed:
- It would be beneficial to clarify how inter-individual variability was accounted for in the study.
- The Authors should provide a more detailed explanation of how this model could be applied in a clinical setting.
Comments on the Quality of English Language
- The Authors are advised to avoid using long and complex sentences to enhance readability.
- Minor typos throughout the manuscript should be corrected for accuracy.
Author Response
Rebuttal Letter
Dear Reviewers,
We sincerely appreciate your thorough review of our manuscript and your constructive feedback. Below, we address each of your comments in detail and outline the corresponding revisions we have made to improve the clarity and scientific rigor of our work.
Reviewer 1
- Clarification on Inter-Individual Variability
Reviewer Comment: "It would be beneficial to clarify how inter-individual variability was accounted for in the study."
Response: We have now explicitly stated in the Methods section under "Animal Model" that:
"Independent of genotype, all mice were female, 10-12 weeks of age, and weighed 20 to 25g at the beginning of the experiments to prevent bias due to inter-individual variability."
This ensures transparency in our approach to standardizing the experimental conditions.
- Clinical Relevance of the Model
Reviewer Comment: "The authors should provide a more detailed explanation of how this model could be applied in a clinical setting."
Response: We have revised the Introduction to better highlight the potential clinical applications of our findings. Specifically, we now state:
"A possible effect of altered microglial cell migration on the development of radiation retinopathy could serve as a novel therapeutic approach for preserving visual function after proton irradiation of the ocular globe."
Furthermore, in the Discussion section, we emphasize the translational relevance:
"The protective role of Cx3cr1 deficiency highlights the need to explore strategies that selectively modulate microglial activity and migration without compromising physiological functions in retinal homeostasis. Given the limited therapeutic options currently available, these findings provide a foundation for developing targeted strategies to preserve vascular integrity, modulate glial and inflammatory responses, and ultimately improve outcomes for patients undergoing proton irradiation. Future research should focus on the mechanistic role of the Cx3cr1 pathway and explore pharmacological interventions translatable to clinical settings, such as intravitreal injections of antibodies against Cx3cr1."
Reviewer 2
- Introduction and choice of the k.o. model
Reviewer Comment: The Introduction is too short, and does not establish a rationale of why the particular knockout mice were chosen. Some of the information included under animal model could be incorporated into the rationale under Introduction.
Response: We appreciate this suggestion. As noted in our response to Reviewer 1, we have already revised the introduction to better explain the rationale for selecting the knockout mice. Additionally, we have clarified the focus of our study on microglia and inflammatory components while referencing existing literature on endothelial cell changes. These modifications have been marked in the revised manuscript (highlighted in yellow).
- Separation of Methods and Results
Reviewer Comment: "The authors need to clearly separate the Methods from the Results, particularly in the section on 'Vascular Component and Gliosis' (Page 8)."
Response: We have carefully revised this section to ensure that methodological descriptions are confined to the Methods section while maintaining clarity in the Results. Where necessary, we have briefly referenced the methodology at the beginning of each results subsection to enhance readability, but without redundant repetition.
- Endothelial Cell Changes
Reviewer Comment: "The authors do not discuss endothelial cell changes post-irradiation."
Response: We acknowledge this point and have included an additional paragraph in the Introduction explaining why endothelial changes were not the primary focus of this study. While endothelial cell loss has been well-documented in previous studies, our research focused primarily on microglia and the inflammatory response, as less is known about their role in this context. Additionally, due to strict regulatory constraints and the limited availability of proton irradiation slots, we had to prioritize specific experimental directions. This is now clarified in the revised Introduction.
“… Since these are small tumor findings, only a few millimeters in size, a focused radiation with minimizing damage to surrounding tissues is desirable.
In everyday clinical practice, plaque brachytherapy with, for example, Ruthenium-106 (beta radiation), Iodine-125 and Palladium-103 (both photon radiation) plaques and, where appropriate facilities are available, external beam radiotherapy with charged particles such as protons have become established [1]. ….
To date, there are no experimental studies using proton irradiation on mouse eyes. But in experiments with external beam radiation from rat eyes, including protons, endothelial cell loss, capillary dropout, vessel occlusion, retinal smooth muscle loss, retinal degradation with photoreceptor loss were observed[1].
Most animal model studies focus on the vasculopathy of radiation retinopathy. Less is known about inflammatory mechanisms, but they also appear to play a significant role. In a publication on radiation retinopathy in rat eyes following irradiation with X-rays, they showed an invasion of micoglia and macrophages into the retinal pigment epithelium and that this observation is related to an outer blood retina barrier disruption[11]. …
A further focus was placed on characterizing inflammatory processes, especially the role of microglia. Therefore, transgenic Cx3cr1 mice were included in the experiments. The homozygote Cx3cr1gfp/gfp mice lack the fractalkine receptor, which is mainly expressed by microglia in the brain and retina and has an important role in trafficking microglia cells to and from the subretinal space[12].
A possible effect of the altered microglia cell migration on the development of radiation retinopathy could be a new therapeutic approach in preserving visual function after proton irradiation of the ocular globe.”
- Formatting Issues
Reviewer Comment: "The figure legends require formatting corrections as different font sizes have been used."
Response: We appreciate this observation. The formatting of all figure legends is done by the journal.
- Reference List Update
Reviewer Comment: "The reference list has not been adequately updated to include all cited references from the text."
Response: We have carefully reviewed and updated the reference list to ensure that all citations in the manuscript are correctly included.
We sincerely appreciate the reviewers' time and effort in improving our manuscript. We believe that the revisions have strengthened the study and look forward to your further feedback.
Best regards,
Susanne Wolf
Reviewer 2 Report
Comments and Suggestions for Authors
Unfortunately, significant improvements are still required in order to make this manuscript useful to the reader.
The Introduction is too short, and does not establish a rationale of why the particular knockout mice were chosen. Some of the information included under animal model could be incorporated into the rationale under Introduction.
The authors need to separate Methods from Results. An example is in the section on 'Vascular component and gliosis' Page 8 section. Furthermore, in this section, and later (knockout animals), the authors are silent on endothelial cell changes. In other words, no results are presented for endothelial cell changes ff irradiation.
Formatting of some of the figure legends require attention, as different font sizes have been used.
The reference list has not been adequately updated to include all cited references from the text.
Author Response
Rebuttal Letter
Dear Reviewers,
We sincerely appreciate your thorough review of our manuscript and your constructive feedback. Below, we address each of your comments in detail and outline the corresponding revisions we have made to improve the clarity and scientific rigor of our work.
Reviewer 1
- Clarification on Inter-Individual Variability
Reviewer Comment: "It would be beneficial to clarify how inter-individual variability was accounted for in the study."
Response: We have now explicitly stated in the Methods section under "Animal Model" that:
"Independent of genotype, all mice were female, 10-12 weeks of age, and weighed 20 to 25g at the beginning of the experiments to prevent bias due to inter-individual variability."
This ensures transparency in our approach to standardizing the experimental conditions.
- Clinical Relevance of the Model
Reviewer Comment: "The authors should provide a more detailed explanation of how this model could be applied in a clinical setting."
Response: We have revised the Introduction to better highlight the potential clinical applications of our findings. Specifically, we now state:
"A possible effect of altered microglial cell migration on the development of radiation retinopathy could serve as a novel therapeutic approach for preserving visual function after proton irradiation of the ocular globe."
Furthermore, in the Discussion section, we emphasize the translational relevance:
"The protective role of Cx3cr1 deficiency highlights the need to explore strategies that selectively modulate microglial activity and migration without compromising physiological functions in retinal homeostasis. Given the limited therapeutic options currently available, these findings provide a foundation for developing targeted strategies to preserve vascular integrity, modulate glial and inflammatory responses, and ultimately improve outcomes for patients undergoing proton irradiation. Future research should focus on the mechanistic role of the Cx3cr1 pathway and explore pharmacological interventions translatable to clinical settings, such as intravitreal injections of antibodies against Cx3cr1."
Round 3
Reviewer 2 Report
Comments and Suggestions for Authors
Unfortunately, the authors have not addressed the earlier query about retinal endothelial cell changes. This is important and requires attention.
Page 4 of the manuscript states "To further investigate changes in the vascular component on a cellular level, isolectin B4 and NG2 were used to label the vascular endothelium and pericytes in retinal flat mounts respectively (Figure 2)."
Page 7 of 24 must include a description EC changes.
Pericyte changes cannot be described as the the primary change without providing evidence on endothelial cell changes. You can have pericyte loss in radiation retinopathy, alongside endothelial cell changes. However, the more predominant loss is what's important here.
My question can be put differently to give you perspective: were there surviving endothelial cells in areas of pericyte loss? (see Gardiner TA et al. Radiation and Diabetic Retinopathy: A Dark Synergy. Int. J. Transl. Med. 2023, 3, 120–159. https://doi.org/10.3390/ijtm3010011).
On a related technical issue, the authors have not been clear as to why the cells described are not endothelial cells. That's again important because it is sometimes difficult to distinguish pericytes and endothelial cells in the mouse retina.
Author Response
Reviewer's Comment: Page 4 of the manuscript states "To further investigate changes in the vascular component on a cellular level, isolectin B4 and NG2 were used to label the vascular endothelium and pericytes in retinal flat mounts respectively (Figure 2)."
Response: We appreciate your attention to our description of the staining methods. We have revised the sentence on page 4 (now 5) to clarify the specificity of the stains used: "To further investigate changes in the vascular component on a cellular level, we employed isolectin B4 (IB4) for general vascular labeling and NG2 for specific pericyte identification in retinal flat mounts (Figure 2)."
This adjustment better reflects our methodology and the limitations of IB4 in distinguishing between endothelial cells and other cell types. We believe that the original statement was misleading and therefore now state in the manuscript, that we did not investigated endothelial cells.
Reviewer's Comment: Page 7 of 24 must include a description of endothelial cell (EC) changes. Pericyte changes cannot be described as the primary change without providing evidence on endothelial cell changes. You can have pericyte loss in radiation retinopathy, alongside endothelial cell changes. However, the more predominant loss is what's important here.
Response: We followed your suggestion and have expanded the discussion in the introduction regarding the importance of endothelial cells in radiation retinopathy. As part of our comprehensive assessment, we performed initial evaluations of the retinal vasculature using Fundus imaging and fluorescence angiography. These assessments showed no significant abnormalities or differences between irradiated and non-irradiated eyes in terms of vessel tortuosity, diameter, or leakage, which informed our decision to not further pursue endothelial cell loss in our primary analyses. Although isolectin B4 (IB4) staining was employed, it was primarily used to visualize the overall vasculature without a specific focus on endothelial cells. IB4 is a broad marker that binds to carbohydrate structures on several cell types, including microglia, which limited its utility in specifically assessing endothelial cell loss.
Our primary focus was on the immune components associated with radiation retinopathy. While we recognize the importance of endothelial cells in the vascular integrity of the retina, our experimental design was directed towards understanding immune responses, particularly microglia activation and pericyte dynamics, which are crucial in the progression of radiation-induced retinal damage.
Reviewer's Question: Were there surviving endothelial cells in areas of pericyte loss?
Response: Our study design did not include a direct assessment of endothelial cell viability in the presence of pericyte loss. This would indeed be an important aspect to investigate further and could provide significant insights, as noted in the referenced study by Gardiner TA et al. Future investigations could benefit from integrating specific endothelial cell viability assays to explore the relationship between endothelial cell survival and pericyte loss.
The manuscript initially reported no significant differences in the basic structure of the vasculature between irradiated and non-irradiated eyes, suggesting that endothelial cells might not have been predominantly affected at the dosages we applied. However, this does not rule out functional changes or subtle damage that could be undetected without specific functional assays. To address your specific query regarding whether there were surviving endothelial cells in areas of pericyte loss, our study design did not include a direct assessment of endothelial cell viability in the presence of pericyte loss, which we now state in the manuscript.
Reviewer's Comment: On a related technical issue, the authors have not been clear as to why the cells described are not endothelial cells. That's again important because it is sometimes difficult to distinguish pericytes and endothelial cells in the mouse retina.
Response: We recognize the challenge in distinguishing between pericytes and endothelial cells in mouse retina studies. In our manuscript, we clarified our staining techniques in the methods section to address this issue. IB4, used for general vascular labeling, has limitations in specificity, which is why we did not solely rely on this staining to assess endothelial cell loss. NG2 was specifically used to label pericytes, and these findings were clear. We acknowledge that additional endothelial-specific markers would provide a more definitive identification and will consider this for future studies.
Round 4
Reviewer 2 Report
Comments and Suggestions for Authors
The revised manuscript represents a significant improvement on the previous versions, and conclusions are supported by the data presented and appropriate interpretations